# Online Geometric Calibration of a Hybrid CT System for Ultrahigh-Resolution Imaging

Dakota H. King [1], Muyang Wang [1], Eric E. Bennett [1], Dumitru Mazilu [1], Marcus Y. Chen [2] and Han Wen [1,*]

1   Biochemistry and Biophysics Center, Laboratory of Imaging Physics, National Heart, Lung and Blood Institute, Division of Intramural Research, National Institutes of Health, Bethesda, MD 20892, USA
2   Cardiovascular CT, Cardiovascular Branch, Division of Intramural Research, National Heart, Lung and Blood Institute, National Institutes of Health, Bethesda, MD 20892, USA
*   Correspondence: han.wen@nih.gov; Tel.: +1-301-496-2694

**Abstract:** A hybrid imaging system consisting of a standard computed tomography (CT) scanner and a low-profile photon-counting detector insert in contact with the patient's body has been used to produce ultrahigh-resolution images in a limited volume in chest scans of patients. The detector insert is placed on the patient bed as needed and not attached. Thus, its position and orientation in the scanner is dependent on the patient's position and scan settings. To allow accurate image reconstruction, we devised a method of determining the relative geometry of the detector insert and the CT scanner for each scan using fiducial markers. This method uses an iterative registration algorithm to align the markers in the reconstructed volume from the detector insert to that of the concurrent CT scan. After obtaining precise geometric information of the detector insert relative to the CT scanner, the two complementary sets of images are summed together to create a detailed image with reduced artifacts.

**Keywords:** CT; surface detector; photon-counting detector; tomosynthesis; hybrid CT; geometric calibration; ultrahigh resolution; image-based calibration

## 1. Introduction

### 1.1. The Hybrid CT Scanner

The hybrid CT scanner consists of a standard CT scanner and an additional low-profile photon-counting detector (hereafter called the "contact detector insert", or CDI) placed on the patient bed in contact with the patient's chest (Figure 1). During a scan, the CT scanner's detector in the rotating gantry and the stationary CDI acquire data simultaneously. The hybrid system allows for ultrahigh-resolution imaging in a limited volume in front of the CDI, as the penumbra of the X-ray source on the CDI is minimized by its close proximity to the patient's body. The design, operation and testing of the hybrid system are described in detail elsewhere [1,2]. The focal spot penumbra is usually the limiting factor of resolution in body scans in current clinical scanners [1]. This is because high levels of X-ray power are required to achieve short scan times, resulting in finite focal spot size. Using this hybrid method, spatial resolution of 150 μm has been demonstrated in a standard CT scanner [1,2].

In the rest of the paper, images reconstructed from the CDI-acquired data are referred to as "CDI images", and the concurrent images provided by the CT scanner itself are referred to as the "CT images". The CDI portion operates in the tomosynthesis mode, where the detector captures a limited range of projection angles for any point in the body in the same way as a digital breast tomosynthesis scanner [3–5].

### 1.2. The Need for Online Geometric Calibration

Geometric calibration is motivated by two primary factors: tomosynthesis reconstruction of the CDI data, and the ability to accurately fuse the reconstructed volumes from the CDI and the CT scanner. Fusion of the images from the two sources requires alignment

of the two reconstructed volumes (Figure 1C). Additionally, tomosynthesis reconstruction of the CDI data requires knowing the position of the X-ray focal spot relative to the CDI at every time point. Both aspects depend on knowing the geometry of the CDI in the CT scanner's bore. Errors in this information will result in misalignment artifacts in the CDI images, and disagreement in the spatial position of features in the CDI images versus the CT images. As the position and orientation of the CDI vary with the patient's weight distribution and posture on the patient bed, they need to be determined for each scan, that is, online calibration is necessary.

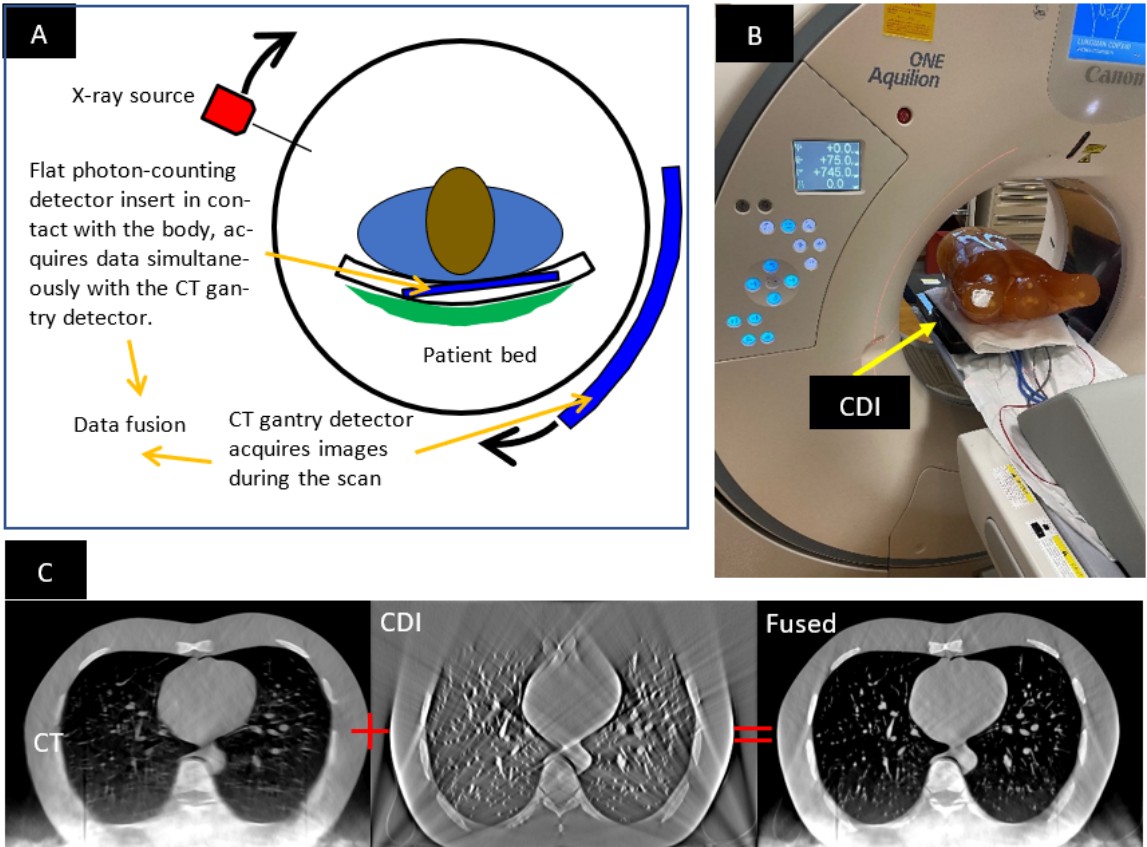

**Figure 1.** (**A**) Schematic of the hybrid imaging system. (**B**) Experimental hybrid system in a commercial CT scanner with the LUNGMAN chest phantom placed on the contact detector insert. (**C**) The images of the hybrid system are a weighted sum of the images from the CDI and the CT scanner. Each CDI and CT image contains missing information and artifacts from truncated or blocked projection angles. They are complementary to each other, such that the fused image has fewer artifacts and more complete depiction of the anatomical features.

### 1.3. What Is New about the Present Geometric Calibration Method

The hybrid CT system has a unique architecture and presents a unique geometric calibration problem, with a unique set of information that can be used for calibration. Previously, a number of calibration methods have been developed for cone-beam tomosynthesis as well as computed tomography to meet the demand of diverse applications [6–41]. They have been generally classified into offline phantom-based calibration methods and online phantom-less methods [28]. Examples of recent literature reviews are provided by Graetz [39] and Jiang et al. [34]. In offline methods, the geometry of the imaging system is determined with dedicated calibration scans of phantoms. The phantoms contain distinct positional markers, such as radio-opaque beads [19,22,23,30,31,35,36,41] or wires [33]. Some phantom-based methods do not need to know the precise arrangement of the markers, as long as the same phantom is scanned in multiple orientations and posi-

tions [9,11,23,24,36,39]. Online calibration methods are used for CT scans with projection angles spanning 180 to 360 degrees. They are phantom-less [7,16–18,20,28,29]. These methods can rely on quantifying misalignment using qualities of the reconstructed image itself: artifacts and image blurring that occur from error in geometric parameters [17,20]. Other methods rely on exploiting the symmetry in a full 360-degree scan [7,16,28].

However, existing online calibration methods are designed for CT scans that cover the full projection range of 180 or 360 degrees. In contrast, data acquired by the CDI have a truncated range of projection angles. It means that in addition to artifacts that are associated with calibration errors, there are artifacts arising from the truncation of the projection angles, making it difficult to isolate and quantify the effect of geometric calibration. This difficulty is mitigated by the fact that in the hybrid system, the commercial CT scanner is fully calibrated, and the CDI detector itself is visible in the CT images. Therefore, we designed a novel method custom to the needs of this system. It is an online method with characteristics of a phantom-based method. Fiducial point markers are placed around the outside of the patient's body, in the z-range covered by the CDI. The true positions of the markers are determined for each scan from the CT images. This then allows us to construct and minimize a cost function that measures the difference between fiducial marker positions in the CDI and the CT images.

## 2. Materials and Methods

### 2.1. Imaging Procedure and the Anthropomorphic Phantom

The hybrid system consisted of a low-profile photon-counting detector as the CDI inside a commercial clinical CT scanner (Canon Aquilion ONE Genesis SP, Canon Medical Systems USA, Tustin, CA, USA). The CDI has a 512 × 4100-pixel matrix (100 um pixel dimension) and can acquire data at a rate of 2000 frames/s at a single energy threshold of 25 keV. The start of the CDI data acquisition was synchronized to the switch-on of the X-ray beam in the CT scanner via the external light indicator of the beam. The CT scan parameters were 120 kV/500 mA, axial (static bed) mode with a single rotation of 1.5 s period and a detector collimation of 120 mm × 0.5 mm. The images from the CT scanner were generated by its commercial software with the manufacturer-recommended chest CT setting.

A Kyoto Kagaku PH-1 LUNGMAN chest phantom (https://www.kyotokagaku.com/en/products_data/ph-1_01/) (accessed on 30 September 2022) was imaged. The LUNGMAN phantom is anthropomorphic in that it simulates an adult human chest with detailed structures of synthetic soft tissue, lung vasculature and bone, each with X-ray absorption comparable to that of real tissue. Fiducial markers were seven tungsten carbide beads of 1 mm diameter. They were attached to a plastic strip in a linear arrangement, which was wrapped around the outer surface of the phantom (Figure 2A). The precise arrangement of the beads was not critical, as long as they were all visible in the CDI images (Figure 2B).

### 2.2. Initial Estimation of the Geometric Parameters

The geometric parameters to be determined are the position and orientation of the CDI in the CT scanner's world coordinate system, and the angle of the X-ray focal spot around its circular trajectory at each time point of the scan.

A method for finding an initial approximation of these parameters was to estimate the position of the CDI inside the CT scanner bore based on CT images that included the CDI. The estimate was imprecise, because the CT images had strong metal artifacts from the CDI's metal parts (Figure 2C). The second part of the initial estimation was the angle of the X-ray focal spot at each time point, based on determining the "sunrise" and "sunset" frames in the projection images acquired by the CDI. These were the time points where the X-ray focal spot aligned with the CDI's image plane, either on its rise above the image plane or fall below the image plane. They were determined visually and imprecise due to several factors, including scattered X-rays that blurred the transition of the image intensity at sunrise and sunset, the finite frame rate of the CDI and the finite focal spot size. These

provided initial estimates of the transformation matrices between the three coordinate systems and the angle offset of the X-ray focal spot at the start of the scan.

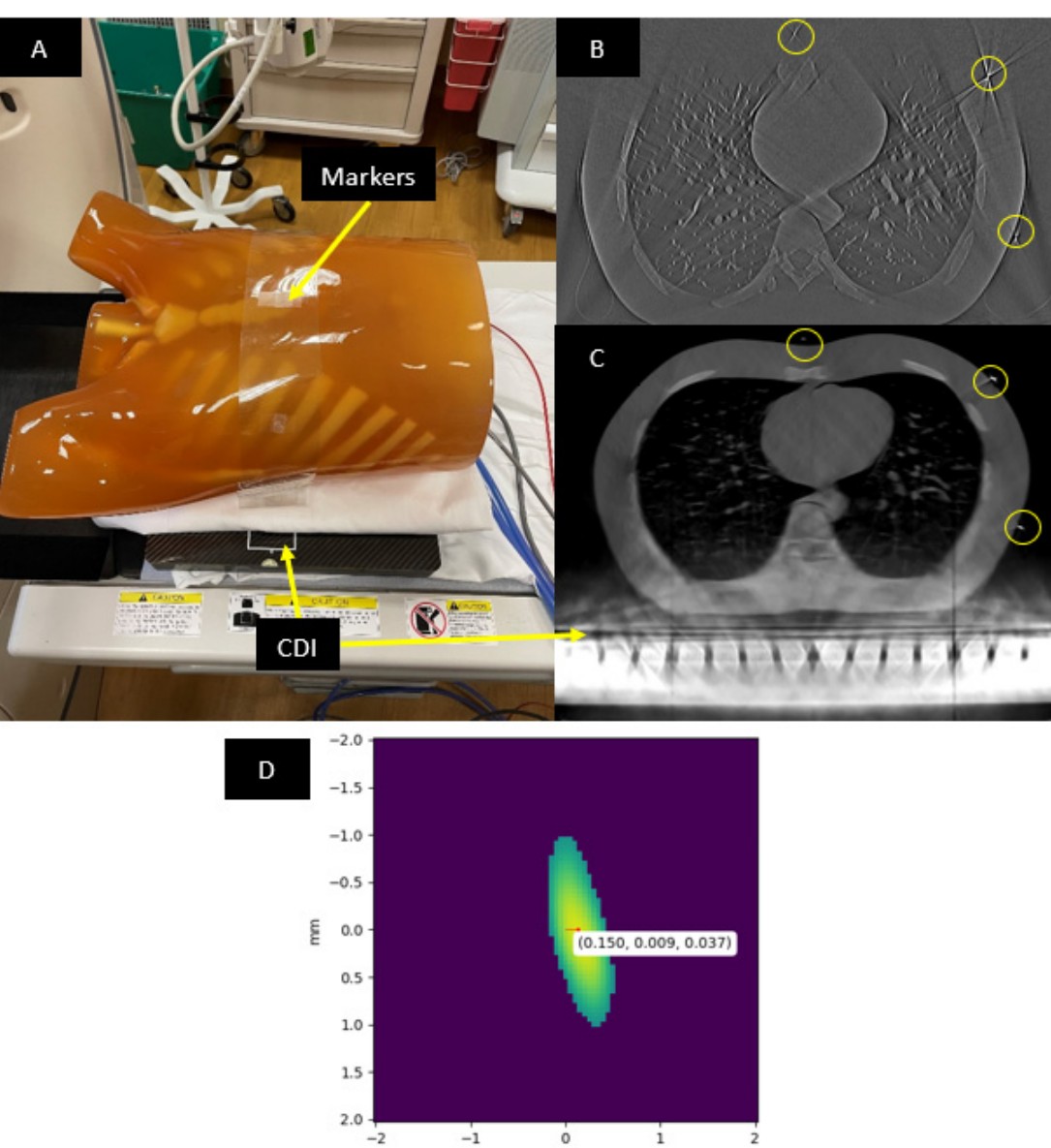

**Figure 2.** (**A**) The LUNGMAN phantom lying on the CDI with fiducial markers placed on the outer surface of the phantom. (**B**) Slice of the CDI's reconstructed volume in the CT coordinate system. (**C**) The same slice in the CT images. Regions circled in yellow highlight visible fiducial markers. (**D**) Intensity profile of a marker bead in a CT image, after thresholding at half maximum intensity. The image was reconstructed at 50 μm pixel size. Center-of-mass of the profile is calculated as the location of a fiducial marker.

### 2.3. Locating the Fiducial Markers in the CT Images

For finding positions of the fiducial markers in the CT images, 25.39 mm field-of-view (FOV) sub-volumes (50 um pixels) with a 0.25 mm slice thickness were reconstructed around each bead using vendor-provided software and settings for chest CT on the scanner console. Each sub-volume was analyzed to find the marker position. First, the region around the marker was segmented out by intensity thresholding at the level of half the maximum intensity (Figure 2D). The segmentation only kept voxel values corresponding to the 3D intensity profile of the fiducial marker. Second, the center-of-mass of this segmented

profile was computed to find the position of the fiducial marker. Third, the position coordinates were transformed to the scanner's world coordinate system using DICOM header information of the sub-volume. These fiducial marker positions were used as the "true" positions to align the CDI's reconstructed volume. To find the positions of markers in the CDI's reconstructed volume, the intensity thresholding and center-of-mass were computed in the same way as with the CT images. The whole process takes 7 min to complete.

### 2.4. Iterative Calibration Algorithm

The calibration concerns three cartesian coordinate systems: the CT scanner's world system, the scanner's image system and the CDI's coordinate system. The CDI's coordinate system is attached to the image matrix of the CDI. The relation between the CT's world and image systems is dependent on the patient's orientation and position in the scanner (https://www.slicer.org/wiki/Coordinate_systems) (accessed on 3 October 2022) according to the convention of the scanner manufacturer. CT scanners provide this information in the DICOM header of their image series. However, their relationship to the CDI's coordinate system needs to be determined for each scan. Additionally, the X-ray focal spot travels on a circular trajectory in the CT world system at a known radius and angular speed, but with an unknown angle offset at the start of each scan. Therefore, the calibration method is aimed at finding 7 parameters, $\vec{p}$ (3 rotational; 3 translational) of the $4 \times 4$ transformation matrix, $T_{CT,\,CDI}$, that maps a 3D point in the CDI system to that of the CT image system, and the angle offset of the X-ray focal spot at the start of the scan, $\varphi_0$.

Starting from the initial estimates $T_{CT,\,CDI}$ and $\varphi_0$ which are described in the previous section "Initial estimation of the geometric parameters", the trajectory of the X-ray focal spot was transformed into the CDI's coordinate system. The projection images from the CDI were then used to reconstruct sub-volumes containing the marker beads via weighted filtered back-projection (WFBP). Using the transformation matrix $T_{CT,\,CDI}$, these sub-volumes were defined in the CT image system and had the same exact voxel positions as the reconstructed sub-volumes from the CT scanner (see previous section "Locating the fiducial markers in the CT images"). This allows a cost function to be created that quantifies the amount of errors in the alignment of the bead positions from the two sources:

$$C(\vec{p},\, \varphi_0) = \frac{1}{N} \sum_{i=1}^{N} \left| \vec{r}_{CT,i} - \vec{r}_{CDI,i} \right|^2, \tag{1}$$

where $\vec{r}_{CT,i}$ and $\vec{r}_{CDI,i}$ are the positions of the marker beads in the CT scanner's reconstructed volume and the CDI's reconstructed volume, respectively. Each of the 7 parameters was then altered in turn, and the WFBP reconstruction of the CDI data was repeated to update the cost function. The process iterated multiple times through the 7 parameters until the cost function C was minimized via Powell's method [42]. The pre- and post-iteration cost function values were compared to quantify improvement in the alignment of the beads.

To reduce the amount of time required for each CDI reconstruction iteration, the multiprocessing and multithreading packages in Python 3.8 were used. Projection images were split into groups where one Python process was responsible for one group. Upon completion of one group, depending on how many groups were needed to compute, the process would continue to work on other groups. In addition to running multiple processes, each process made use of multiple threads to further reduce computation time. These measures reduced the total time of the process from about 450 h to 13 h.

### 2.5. Evaluation of the Calibration Method Using Blood Vessels

For additional quantification of alignment, 2D positions of small vessel structures from the CDI and CT images were compared in several image slices spanning the reconstructions. Stacks of 240 images of 101.6 FOV (0.198 mm pixel size) and 0.25 mm slice thickness were reconstructed. Within a slice, contours were defined around each vessel using a

threshold of half the maximum intensity of the vessel; the center of mass of the pixel values within the contour was used to calculate the position. Small vessels, 2 mm or smaller in diameter, were chosen for this analysis. Ten vessels were analyzed in total: three located in slices in the +z direction, three in the -z direction and four near the middle slice. In addition, the vessels chosen were distributed in different regions in the pixel grid (x- and y-coordinates) to represent various regions of the phantom. Pre- and post-calibration errors were compared to give an additional quantification and verification of alignment throughout the reconstructed volumes.

### 3. Results

The calibration algorithm completed minimization of the cost function in 120 iterative steps based on the criteria of Powell's method [42]. Figure 3 summarizes the quantitative results of the iterative procedure. The root-mean-squared (RMS) of the misalignment of the fiducial marker positions between the CDI and CT images was reduced by 68% with the procedure, from 0.563 mm to 0.182 mm (Figure 3A). As a result, the RMS of the 2D misalignment of small blood vessels in the axial slices was reduced by 65%, from 0.375 mm to 0.131 mm (Figure 3B). An example of improved alignment of small vessels is shown in Figure 3C.

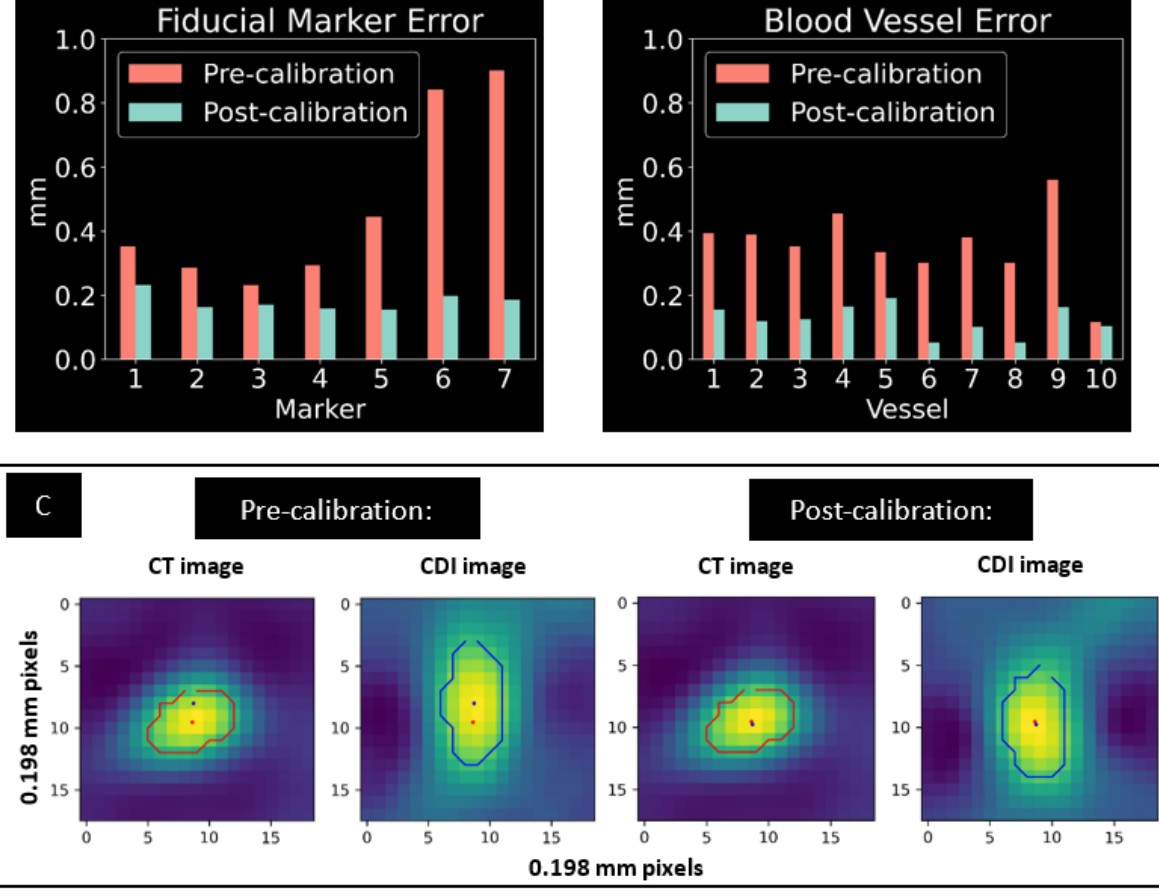

**Figure 3.** (**A**) Pre- and post-calibration 3D alignment errors of the marker beads between the CDI images and the CT images. (**B**) Pre- and post-calibration 2D alignment errors of small blood vessels (< 2 mm diameter) between the two detectors in the axial plane. The vessels were distributed evenly in the reconstructed volume. (**C**) An example of a blood vessel position in the CT images (red contour) and CDI images (blue contour). The red and blue dots are the center-of-mass positions of the vessel in the CDI and CT images, respectively.

The overall effect of the iterative calibration procedure on the reconstructed images of the hybrid system is illustrated in magnified regions of the right and left lungs (Figure 4). Fusion of the two sources of data with pre-calibrated parameters resulted in substantial streaking artifacts and misalignment of features such as vessel segments from the two sources (Figure 4B). After the iterative procedure, the streaking artifacts were reduced and the vessel segments were aligned (Figure 4C).

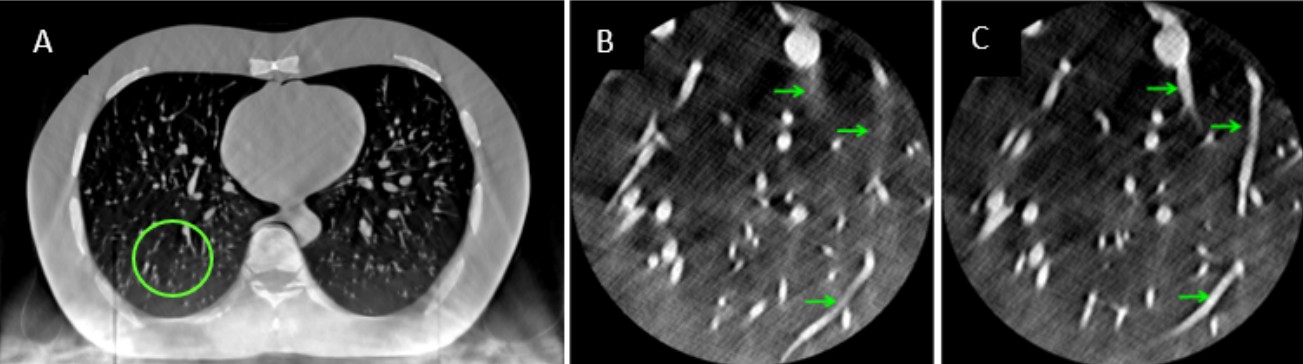

**Figure 4.** (**A**) A 500 mm FOV image of the chest phantom after data fusion. (**B**,**C**) Pre- and post-calibration 50 mm FOV (0.098 mm pixel) CT–CDI fused images of a region highlighted by the green circle in (**A**). Arrows point to the difference in the visualization of vessel structures before and after the iterative calibration.

## 4. Discussion

The hybrid CT system presented a unique geometric calibration problem and required on-line calibration for each scan. The method presented here is a hybridization of phantom-based calibration and image-based iterative optimization. As verified by fiducial marker and blood vessel alignment, the method was able to determine the geometric parameters of the system on a per-scan basis. Starting from an initial estimate of the geometric parameters based on the appearance of the CDI detector in the CT images, the process reduced the misalignment between the CDI and CT images by 65%, to between 0.1 and 0.2 mm. The resulting image from the fusion of the two datasets provided a more complete representation of the object, as the two datasets each contain complementary information.

Limitations of this calibration method arise from the statistical error in locating fiducial marker positions in both the CT and CDI reconstructed volumes. The locations of the markers depended on the threshold value applied to the image before calculating the center of mass, the resolution of the reconstruction in all three dimensions and the presence of image artifacts. The calculated marker positions fluctuated by about 0.1 mm between different threshold values. This level of uncertainty and the errors associated with tomosynthesis artifacts can explain the residual misalignment of 0.1 to 0.2 mm between the CDI and CT datasets after the iterative process. Choosing tungsten carbide as the fiducial marker material helped to retain the majority of the marker intensity profile after thresholding the image due to the markers' high relative brightness. However, choosing the threshold value is somewhat subjective and can lead to slightly different results; we found that using a consistent half-of-maximum intensity value was adequate to achieve desired calibration results in this initial study.

Further improvements of this calibration method will aim to shorten the total computation time required. As an online method, geometric calibration will need to be performed for each patient scan, and it would be ideal to reduce the total computation time down to seconds. This goal requires further code optimization in addition to larger parallel processing capability. To improve alignment results, image artifacts in the CDI's reconstruction need to be reduced with refined reconstruction algorithms. Suppressing these artifacts may improve the intensity profile of the fiducial markers via reduced streaking, and in turn improve the precision of the calibration.

**Author Contributions:** Conceptualization, H.W.; methodology, D.H.K. and H.W.; software, D.H.K. and H.W.; validation, D.H.K., M.W. and H.W.; formal analysis, D.H.K. and H.W.; resources, E.E.B., D.M., M.Y.C. and H.W.; data curation, D.H.K.; writing—original draft preparation, D.H.K.; writing—review and editing, H.W.; visualization, D.H.K.; supervision, H.W.; funding acquisition, H.W. All authors have read and agreed to the published version of the manuscript.

**Funding:** This research was funded by the Division of Intramural Research, National Heart, Lung and Blood Institute, Intramural Research Program, National Institutes of Health, USA, project number HL006141-12.

**Institutional Review Board Statement:** Not applicable.

**Informed Consent Statement:** Not applicable.

**Data Availability Statement:** Not applicable.

**Acknowledgments:** We are grateful to Shirley Rollison of Radiology and Imaging Sciences Dept. of NIH Clinical Center for her assistance with the CT scans.

**Conflicts of Interest:** The authors declare no conflict of interest.

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
