# Peer review of "Online Geometric Calibration of a Hybrid CT System for Ultrahigh-Resolution Imaging"

_tomography, doi:10.3390/tomography8050212_

Round 1

Reviewer 1 Report

Comments to the Author

Title: Online Geometric Calibration of a Hybrid CT System for Ultra-high-Resolution Imaging

Authors: Dakota H. King, Eric E. Bennett, Dumitru Mazilu, Muyang Wang, Marcus Y. Chen, and Han Wen

à The paper presents a method to determine the relative geometry of a detector insert and a CT scanner using fiducial markers. The proposed method uses an iterative registration algorithm to align the markers in the reconstructed volume from the detector insert to that of the concurrent CT scan. The two complementary sets of images are summed together to create a detailed image with reduced artifacts.

à I believe this is an interesting work however needs to be polished and some more information is needed before being suitable for publication in Tomography. I would consider the publication of the present manuscript after carefully addressing the comments hereafter and modifying its structure. 

General: The manuscript is well written. The use of English is correct as well as the length for each section.

In the following: MA=Major comment, MI = Minor comment, OP = Optional Comment

(MI) P1, Abstract, L10: Since this is the first time appearing in the text, include the meaning of CT: Computed Tomography (CT).

(MI) P1, Introduction, L26: The explanation if CT should appear the first time CT is used, so please, move the explanation at the beginning of the sentence.

(OP) P1, Introduction, L26: Which CT scanner are you using for the experiment?

(MI) P2, Introduction, L33-35: Add references to support this statement.

(OP) P2, Introduction, L40: I would describe what tomosynthesis mode is in a few lines.

(MI) P2, Introduction, Fig 1A: Some of the words are cut or covered by other images/arrows.

(OP) P2, Introduction, L42: Explain what is the geometric calibration accounting for.

(MI) P2, Introduction, L60: CT has been already defined.

(MA) P3, Introduction, L60: References must be sorted in order of appearance. Please rearrange your reference list.

(MA) P3, Introduction, L66-67: Rewrite this sentence, it is confusing.

(MI) P3, Introduction, L72: Use always the same style. You were not using the hyphen at the beginning of this paragraph.

(MA) P3, Introduction, L73: I think this is incorrect, and you mean: However, existing online calibration methods...

(MA) P3, Introduction, L73-84: This paragraph is a bit difficult to understand. Please. Re-work this section.

(MI) P3, Materials and methods, L90: Try to homogenize the styles. Do not write 4100 and the 2,000. Choose one or the other.

(MI) P3, Materials and methods, L94: I think this should be mm^2.

(MI) P3, Materials and methods, L95: Add reference to the phantom.

(MA) P4, Materials and methods, Figure2B: Figure 2B was never introduce in the text. You have to introduce this figure or just remove it.

(MA) P4, Materials and methods, Figure2D: The black numbers inside the figure cannot been read. Change the color or write them in a box.

(MI) P5, Materials and methods, L123: Define FOV.

(MI) P5, Materials and methods, L131: Add reference to the DICOM header.

(MI) P5, Materials and methods, L134: How long does it take the process.

(OP) P5, Materials and methods, L43: Consider including a drawing with the different coordinate systems.

(MA) P5, Materials and methods, L147: This is incorrect. The parameters were not defined in the previous section, at least not using the same letters. Please check.

(MI) P5, Materials and methods, L160: How long does it take the iterative process.

(MI) P5, Materials and methods, L160: How much time did it save using the python processing?

(MI) P6, Materials and methods, L172-173: Mention how many images you acquired. More information is needed regarding the used images.

(MI) P6, Results, L184: How did you decided to use 120 iterations?

(MI) P7, Results, L189: Mention that this is Figure 3C.

(MI) P7, Results, L196: There is no Figure 4E.

(MI) P7, Discussion, L203: Provide the numbers shown in the results section.

(MI) P8, Discussion, L220: Could you briefly mention these improvements?

(MI) P8, Discussion, L681: The authors should provide a closing paragraph describing the future perspective of these techniques.

(MI) P8, Conclusion, L681: The authors should include few lines comparing their results with those provided by other methods

(MA) References: Please, check that all references follow the same style as outlined by the Journal author instructions.

Author Response

We greatly appreciate the detailed and helpful comments provided by the reviewer. The following is a point-by-point reply to the comments. The comments are in quotation marks.

“(MI) P1, Abstract, L10: Since this is the first time appearing in the text, include the meaning of CT: Computed Tomography (CT).”

This change is made.

“(MI) P1, Introduction, L26: The explanation if CT should appear the first time CT is used, so please, move the explanation at the beginning of the sentence.”

This is changed as suggested.

“(OP) P1, Introduction, L26: Which CT scanner are you using for the experiment?”

It is a Canon Aquilion ONE Genesis SP. The information is added to the Materials and Methods section.

“(MI) P2, Introduction, L33-35: Add references to support this statement.”

Reference [1] extensively discusses this point and is added here.

“(OP) P2, Introduction, L40: I would describe what tomosynthesis mode is in a few lines.”

We added a brief description of tomosynthesis.

“(MI) P2, Introduction, Fig 1A: Some of the words are cut or covered by other images/arrows.”

This is corrected.

“(OP) P2, Introduction, L42: Explain what is the geometric calibration accounting for.”

This is explained in detail in section 1.2.

“(MI) P2, Introduction, L60: CT has been already defined.”

This is corrected.

“(MA) P3, Introduction, L60: References must be sorted in order of appearance. Please rearrange your reference list.”

We double-checked this point. Our intention is to first provide a fairly comprehensive list of calibration papers [6-41] in the sentence L60-61, in the chronological order that they were published, as is conventional for reviewing the history of a topic (giving significance to earliest works). Then, In the following text, we descend into several sub topics, and in each we refer to only a select subset of the list [6-41] which are relevant to our work. Some of the papers are cited twice in this structure and some are not.

“(MA) P3, Introduction, L66-67: Rewrite this sentence, it is confusing.”

We corrected the sentence to “Some phantom-based methods do not need to know the precise arrangement of the markers, as long as the same phantom is scanned in multiple orientations and positions”

“(MI) P3, Introduction, L72: Use always the same style. You were not using the hyphen at the beginning of this paragraph.”

This is corrected.

“(MA) P3, Introduction, L73: I think this is incorrect, and you mean: However, existing online calibration methods...

(MA) P3, Introduction, L73-84: This paragraph is a bit difficult to understand. Please. Re-work this section.”

We revised the description to “However, existing online calibration methods are designed for CT scans that cover the full projection range of 180 or 360 degrees. In contrast, data acquired by the CDI have a truncated range of projection angles. It means that in addition to artifacts that are associated with calibration errors, there are artifacts arising from the truncation of the projection angles, making it difficult to isolate and quantify the effect of geometric calibration. This difficulty is mitigated by the fact that in the hybrid system, the commercial CT scanner is fully calibrated, and the CDI detector itself is visible in the CT images.”

“(MI) P3, Materials and methods, L90: Try to homogenize the styles. Do not write 4100 and the 2,000. Choose one or the other.”

We corrected this problem.

“(MI) P3, Materials and methods, L94: I think this should be mm^2.”

It is a standard way to represent CT detector collimation: 120 rows of 0.5 mm each row.

“(MI) P3, Materials and methods, L95: Add reference to the phantom.”

We added a reference to the product.

“(MA) P4, Materials and methods, Figure2B: Figure 2B was never introduce in the text. You have to introduce this figure or just remove it.”

Figure 2B should have been referenced in L104. This is corrected.

“(MA) P4, Materials and methods, Figure2D: The black numbers inside the figure cannot been read. Change the color or write them in a box.”

This is corrected.

“(MI) P5, Materials and methods, L123: Define FOV.”

The definition is added.

“(MI) P5, Materials and methods, L131: Add reference to the DICOM header.”

DICOM is the name of an international standard for medical image meta data for several decades and is conventionally used in papers without references except in works dealing with the standard itself.

“(MI) P5, Materials and methods, L134: How long does it take the process.”

The process takes about 7 minutes. This information is added.

“(OP) P5, Materials and methods, L43: Consider including a drawing with the different coordinate systems.”

The relationship between the CT’s world system and image system is dependent on the patient’s orientation and position in the scanner and the convention of the scanner manufacturer. It is too complex to illustrate in a diagram. Instead, we added a reference to the website https://www.slicer.org/wiki/Coordinate_systems that explained the relationship between the two with detailed graphics.

“(MA) P5, Materials and methods, L147: This is incorrect. The parameters were not defined in the previous section, at least not using the same letters. Please check.”

This is corrected by adding the information to section 2.2.

(MI) P5, Materials and methods, L160: How long does it take the iterative process.

It took 13 hours to complete the 120 iterations. This information is added.

(MI) P5, Materials and methods, L160: How much time did it save using the python processing?

We added that the parallel processing measures reduced the total time of the process from 450 hours to 13 hours.

“(MI) P6, Materials and methods, L172-173: Mention how many images you acquired. More information is needed regarding the used images.”

We added the information “Stacks of 240 Images of 101.6 FOV (0.198 mm pixel size) and 0.25 mm slice thickness were reconstructed.”  

“(MI) P6, Results, L184: How did you decided to use 120 iterations?”

The point at which the iterations converge is based on the criteria of the Powell’s method (ref. [42]). This information is added.

“(MI) P7, Results, L189: Mention that this is Figure 3C.”

Figures 3A, 3B and 3C are now specifically mentioned in the Results section.

“(MI) P7, Results, L196: There is no Figure 4E.”

This section of Results is corrected to match Figure 4.

“(MI) P7, Discussion, L203: Provide the numbers shown in the results section. (MI) P8, Discussion, L220: Could you briefly mention these improvements?”

As suggested a summary of the data in Results is added to the Discussion section.

“(MI) P8, Discussion, L681: The authors should provide a closing paragraph describing the future perspective of these techniques.”

This is last paragraph starting with “Further improvements of this calibration method will aim to ……”

“(MI) P8, Conclusion, L681: The authors should include few lines comparing their results with those provided by other methods”

This is a unique hybrid CT imaging approach with a unique set of information that can be used for the calibration process. As a result, existing methods do not apply to this situation and we needed to design a custom method for this approach. Therefore, there are no existing methods that can be used as a baseline for comparison. This point is added to section 1.3.

“(MA) References: Please, check that all references follow the same style as outlined by the Journal author instructions.”

This is done in the revision.

Reviewer 2 Report

This paper presents a method for geometric calibration between two CT subsystems in a hybrid imaging system consisting of a standard CT scanner and a low-profile photon-counting detector insert in contact with the body of the patient. The contact detector insert offers ultrahigh-resolution imaging in a limited volume in chest scans of patients, however, the approach requires accurate information about the physical position and orientations of the detector relative to the x-ray source and the ordinary CT detector. The geometrical calibration is obtained by scanning of seven fiducial markers attached to an anthropomorphic chest phantom and subsequent minimization of the alignment errors using an iterative algorithm. The approach is straight forward and sound, and convincing results are obtained.

In general the manuscript is very well written and the results are clearly presented, however, selected parts needs clarification.

The work is recommended for publication after minor revision.

Specific comments:

1) In the abstract the combined use of the two complementary set of images is simply described as “summed together”, however, according to the caption of figure 1 caption the “fused” image is calculated as a weighted sum of the images from the CDI and the CT. It seems that the fused image in figure 1C may not be generate as a simple weighted sum, but also must involve some other operations like thresholding?

2) More information about the commercial CT scanner should be provided, e.g. brand, model, detector dimension/pitch.

3) The single energy threshold for the photon-counting CDI should be specified. It may be 25 keV as specified in reference [1], however, this seems to be for a previous version of the CDI unit.

4) The reconstruction algorithm and relevant parameters used for the standard CT images should be specified (probably FBP).

5) For most modern imaging systems (CT, SPECT, PET) FBP has largely been replaced by statistical reconstruction (maximum likelihood), which has shown to be very efficient for reduction of noise and artifacts (e.g. from metal parts in the CDI). This aspect should be commented.

6) The computation time using the current implementation should be specified (e.g. line no. 184).

7) The method id validated using ten vessels located in either +z, -z or central slices. How large an axial FOV is used for this analysis? It is mentioned in line no. 123 that 25 mm FOV is used for the images used to localize the fiducial markers.

8) It is stated that the alignment errors are defined as RMS values. For fiducial markers the position would obviously be well defined in all three dimensions (x,y,z). However, for the vessels extending along one direction, it seems that the position may be defined only in two dimensions. Please claify.

Author Response

Reply to review comments: we are grateful to the reviewer for the helpful comments. Below is a point-by-point reply to the comments. The reviewer comments are in quotation marks.

“Specific comments:

1) In the abstract the combined use of the two complementary set of images is simply described as “summed together”, however, according to the caption of figure 1 caption the “fused” image is calculated as a weighted sum of the images from the CDI and the CT. It seems that the fused image in figure 1C may not be generate as a simple weighted sum, but also must involve some other operations like thresholding?”

The fused image is currently a simple weighted sum of the two sets of images without involving additional calculations. In the future we may consider more sophisticated algorithms to fuse the two sets.

“2) More information about the commercial CT scanner should be provided, e.g. brand, model, detector dimension/pitch.”

This information is now added to the Materials and Methods section.

“3) The single energy threshold for the photon-counting CDI should be specified. It may be 25 keV as specified in reference [1], however, this seems to be for a previous version of the CDI unit.”

It was 25 keV and the information is added to the Materials and Methods section.

“4) The reconstruction algorithm and relevant parameters used for the standard CT images should be specified (probably FBP).”

Unfortunately, we do not have information on the reconstruction algorithms of the CT scanner. The images from the CT scanner were generated by its commercial software with the manufacturer-recommended chest CT setting. The projection images were inaccessible to us, and the reconstruction algorithms were proprietary and not disclosed to us. Some parameters such as field-of-view and slice thickness were set by us and specified in the manuscript.

“5) For most modern imaging systems (CT, SPECT, PET) FBP has largely been replaced by statistical reconstruction (maximum likelihood), which has shown to be very efficient for reduction of noise and artifacts (e.g. from metal parts in the CDI). This aspect should be commented.”

In this work the CDI portion of the reconstruction was still WFBP, and we are working to advance it to modern methods. The images provided by the CT scanner were generated by the scanner’s own software and involved algorithms that were not disclosed to users.

6) The computation time using the current implementation should be specified (e.g. line no. 184).

This is now specified to be 13 hours for the 120 iterations.

7) The method id validated using ten vessels located in either +z, -z or central slices. How large an axial FOV is used for this analysis? It is mentioned in line no. 123 that 25 mm FOV is used for the images used to localize the fiducial markers.

We added the information “Stacks of 240 Images of 101.6 FOV (0.198 mm pixel size) and 0.25 mm slice thickness were reconstructed” for the validation with vessels.

“8) It is stated that the alignment errors are defined as RMS values. For fiducial markers the position would obviously be well defined in all three dimensions (x,y,z). However, for the vessels extending along one direction, it seems that the position may be defined only in two dimensions. Please claify.”

Yes, the vessel misalignment is measured in 2D axial slices. This is defined in section 2.5 and now also specified in Fig. 3 caption.

Round 2

Reviewer 1 Report

Dear authors,

Thanks for considering and addressing my comments and suggestions. Good job